# High endemicity of Q fever in French Guiana: A cross sectional study (2007–2017)

**Pauline Thill** [1,2], **Carole Eldin**[3,4], **Laureen Dahuron**[1], **Alain Berlioz-Artaud**[5], **Magalie Demar**[6,7], **Mathieu Nacher**[8], **Emmanuel Beillard**[5], **Félix Djossou**[1,7], **Loïc Epelboin** [1,7,8]*

**1** Unité des Maladies Infectieuses et Tropicales, Centre Hospitalier de Cayenne Andrée Rosemon, Cayenne, French Guiana, **2** Service Universitaire de Maladies Infectieuses et du Voyageur, Centre Hospitalier Tourcoing, France, **3** Aix-Marseille Université, IRD, AP-HM, SSA, VITROME; IHU-Méditerranée Infection; Centre de référence pour la prise en charge des maladies vectorielles à tiques, Marseille, France, **4** French Reference Center for the Diagnosis and Study of Rickettsioses, Q Fever and Bartonelloses, IHU-Méditerranée Infection, Marseille, France, **5** Laboratoire de Biologie polyvalente, Institut Pasteur de la Guyane, Cayenne, French Guiana, **6** Laboratoire Hospitalo-Universitaire de Microbiologie, Centre Hospitalier de Cayenne Andrée Rosemon, Cayenne, French Guiana, **7** TBIP, Université de Guyane, Cayenne, French Guiana, **8** Centre d'Investigation Clinique Antilles Guyana (CIC Inserm 1424), Centre Hospitalier de Cayenne Andrée Rosemon, Cayenne, French Guiana

* epelboincrh@hotmail.fr

**Data Availability Statement:** Data are available from the Centre d'Investigation Clinique Inserm 1424, Centre Hospitalier de Cayenne, French Guiana Institutional Data Access for researchers

## Abstract

Q fever (QF) is a zoonosis caused by *Coxiella burnetii* (Cb). French Guiana (FG) had a high incidence but no data have been published since 2006. The objective of this study was to update the incidence and epidemiological data on QF in FG. A retrospective study of all FG Q fever serodiagnosis between 2007 and 2017 was carried out. Among the 695 patients included, the M/F sex-ratio was 2.0 and the median age of 45.3 years (IQR 33.7–56.3). The annual QF incidence rate was 27.4 cases (95%CI: 7.1–47.7) per 100,000 inhabitants ranging from 5.2 in 2007 to 40.4 in 2010. Risk factors associated with Q fever compared to general population were male gender, being born in mainland France, an age between 30 to 59 years-old and a residence in Cayenne and surroundings. The incidence of QF in FG remains high and stable and the highest in the world.

## Author summary

We present here a study showing the exceptional nature of the incidence of Q fever in French Guiana. Indeed, this zoonosis due to the bacterium *Coxiella burnetii*, is a real public health problem in French Guiana, a French ultra-marine territory located in the North East of South America. The study found an endemic state with a stable incidence between 2010 and 2017 around 25–30 cases per 100,000 inhabitants per year. More than 90% of cases are concentrated in the territory's capital, Cayenne, and its surroundings. The risk factors for Q fever are being male, being between 30 and 59 years old, which are risk factors found elsewhere, but also living in Cayenne and its surroundings and being born in mainland France or Europe.

who meet the criteria for access to confidential data via the following email address: publication. recherche@ch-cayenne.fr.

**Funding:** The authors received no specific funding for this work.

**Competing interests:** The authors have declared that no competing interests exist.

## Introduction

Q fever (QF) is a worldwide bacterial zoonosis caused by *Coxiella burnetii* (*Cb*). *Cb* is a small, obligate intracellular pathogen and is endemic worldwide in a variety of wild and domestic mammals, and arthropods [1]. The diagnosis of acute Q fever mostly relies on the detection of anti-*C. burnetii* antibodies 15 to 21 days after the symptom onset by Immuno-Fluorescence Assay (IFA), which is the gold standard for *C. burnetii* antibodies detection.

In most developed countries, QF infection has been extensively described. The physiopathology, exposure factors, and clinical and biological presentations of the disease are well established. The incidence of Q fever is generally quite low (<2 case/100,000 inhabitants per year) [2], and most of the cases are diagnosed during short outbreaks related to direct or indirect contacts of humans with goat and sheep, which are the main reservoirs. In developing countries however, information on endemicity is limited to seroprevalence studies in exposed populations or case reports [1].

French Guiana is a French overseas territory located on the northeastern coast of South America. About 90% of its 84,000 km$^2$ surface is covered by the Amazonian rainforest [3]. The remaining 10% is a coastal plain where 90% of the 271,829 inhabitants live (INSEE (The national institute of statistics and economic studies) 2019 www.insee.fr). The cycle of seasons in French Guiana is punctuated by the variations in latitude of the Inter Tropical Convergence Zone which determines 4 seasons: small rainy season from December to mid-February. Small summer of March between mid-February and the end of March (very variable from one year to another). Big rainy season in April, May and June. Tropical wave period in July and August (intermediate period between the great rainy season and the dry season). Dry season: from September to the end of November. Almost half of the population lives in Cayenne, the capital city of FG, and its surroundings, including Rémire-Montjoly and Matoury. The main cities are Cayenne (60,947 inhabitants), Kourou (26,726 inhabitants) and Saint Laurent du Maroni (44,130 inhabitants) (INSEE Census 2019; URL: http://www.insee.fr). The healthcare offer in French Guiana is divided between 3 public hospitals located in the three main cities of the territory (Cayenne, Kourou and Saint Laurent du Maroni), as well as 18 health centers spread throughout the territory, including in the most remote areas. In 2020, 648 physicians were registered with the Departmental Council of the Order of Physicians of French Guiana, of which 583 were active [4].

*Cb* was first described in French Guiana in 1955 in cattle and 1957 in humans [5,6]. A subsequent report was published in1998 with the description of three severe cases [7]. Previous published data also showed that QF in FG has different clinical and biological signs than in Europe: patients with QF are more symptomatic, more often febrile in Cayenne than in Marseille [8]. The most frequent clinical presentation of Q fever in FG is pneumonia which is observed in 90% of the cases), at a much higher proportion than in mainland France (40%). *Cb* is implicated in 24% to 38% of community-acquired pneumonia in Cayenne while this pathogen is barely involved (<1%) in continental Europe and in the rest of the world [9]. In addition, a single clone has been shown to be involved in both human and animal Q fever in French Guiana, which strain Cb175 MST 17 is thought to be more virulent than the European reference strains, and is not found anywhere else in the world [10–12]. Concerning the incidence of QF in FG, a significant increase in the incidence rate was observed in 1996 [7,13]. During the 1996–2000 period, the annual incidence of Q fever in French Guiana was very high (37 cases per 100,000 inhabitants per year) and increased to 149.9 cases per 100,000 inhabitants per year in 2005 [14,15]. The incidence of Q fever in FG thus seems to be one of the highest in the world, but no data about epidemiological features and incidence have been updated since 2006 [15]. The primary aim of our study was to update the incidence data and epidemiological

features of Q fever in FG with the most recent results available. The secondary objectives were to study the sociodemographic characteristics of the population, determine risk factors of QF in FG, and to determinate if climate seasonal factors and geographic distribution had any influence on this disease onset [16].

## Methods

### Ethics statement

This is a non-interventional study of retrospective data collected only in the context of routine care. The data were collected from a single center, the Groupement Hospitalier de Territoire de Guyane (Territory Hospital Grouping of French Guiana), which includes the 3 coastal hospital centers (Saint Laurent du Maroni, Kourou, and Cayenne) and the 17 health centers of the remote area of French Guiana, that are administratively dependent on the Cayenne Hospital Center. Some patients have been diagnosed by a general practitioner, but were then referred to the specialized consultation at the hospital of Cayenne. Data collection in the medical record was done pseudonymously, for all patients diagnosed with Q fever between 2007 and 2017. Only the information strictly necessary for the treatment and purpose of the research was collected and these data kept for the duration of the study and then archived for 15 years. The data were identified by a code and the patients' initials. Finally, the data were collected from data already collected previously from medical records. This research was in compliance with the law "Informatique et Libertés" of January 6, 1978 as amended and the law No. 2018–493 of June 20, 2018 on the protection of personal data and Regulation (EU) 2016/679 of the European Parliament and of the Council of April 27, 2016 on the protection of individuals with regard to the processing of personal data and on the free movement of such data (RGPD). The data was transferred and collected in accordance with the reference methodology MR004 of the Commission Nationale de l'Informatique et des Libertés (CNIL) for which the Centre Hospitalier de Cayenne has signed a compliance commitment.

### Study design and population

We conducted a retrospective multicenter study including every patient with Q fever diagnostic testing in French Guiana from Jan. 1$^{st}$2007 to Dec. 31$^{st}$2017.The study population included any patient with a positive serodiagnosis of Q fever performed in the Institut Pasteur de la Guyane, in the Q fever National Reference Center in Marseille, France, and in Biomnis or Cerba central laboratories and patients admitted in Cayenne hospital with a diagnosis of Q fever. The Biomnis laboratory is located in Ivry-sur-Seine and the Cerba laboratory in Saint-Ouen-l'Aumône, both in the Paris suburbs.

### Inclusion and exclusion criteria and case definition

The inclusion criteria were based on cases with compatible clinical picture associated with a positive serodiagnosis of *Cb* infection. The diagnosis relied on serology using immunoglobulin G (IgG) and immunoglobulin M (IgM) antibodies against phase II (acute infection) and phase I (chronic infection) antigens of C. burnetii in an indirect immunofluorescence assay [1,17]. ». All four laboratories used the same criteria for a positive test, criteria that were issued by the French National Reference Center of Q fever in Marseille [17]. A specimen was considered positive if antibody titers against phase II *C. burnetti* were above 200 (IgM) or 400 (IgG). The diagnosis of acute Q fever relied on clinical criteria (fever, hepatitis or pneumonia over the 3 preceding months) and/or positive phase II anti-*Cb* IgG to 400 or IgM superior to 200, or IgM seroconversion from negative to ≥ 50. Persistent focal Q fever (former chronic Q fever) relied

on clinical criteria (presence of endocarditis, vascular lesions, or any compatible clinical picture) and/or IgG antibody titers against phase I *C. burnetti* above 800. Patients with a first diagnosis of acute Q fever secondarily developing a persistent form were included as acute forms. Patients were excluded if no medical data was available. Because of the high seroprevalence of Q fever in French Guiana, patients with a clinical picture not very suggestive of a picture, and a phase 1 or 2 IgG serology greater than or equal to 200 without IgM were considered as serological scars and were secondarily excluded.

### Data collection and analysis

Medical records were reviewed to confirm if the disease was acute or persistent. Demographic data (gender, age, place of residence (town and neighborhood) and country of birth), and climate data (dry or wet season) were collected retrospectively. The patients with acute Q fever were classified into 3 categories: pneumonia, isolated fever, other acute Q fever. Patients with a compatible serological profile, with only "acute Q fever" in the medical chart but without details about the clinical picture were included anyway and classified as "no clinical data". Patients with persistent clinical pictures were classified into 4 categories: endocarditis, vascular infection, dilated cardiomyopathy, and "other form".

In order to enhance specificities of patients with Q fever, these data were compared to those of the global population of French Guiana: the 2015 population census data (URL: http://www.insee.fr) using Fisher's exact test.

Epidemiologic curves were built using annual incidence rate during the study period, 2007–2017 and completed with incidence data from 1990 [15].

### Calculation of the seropositivity rate

In a second step, in 2022, we sought to calculate the rate of *C. burnetii* serology positivity. We therefore contacted the 4 laboratories where the samples came from to find out the number of patients tested for Q fever during the study period. The total number of patients in the study with a diagnosis of Q fever was calculated by dividing the number of patients tested for Q fever by the number of patients with a non-zero IgM and/or IgG titre against *C. burnetii*. A map showing the distribution of cases according to the districts over the study period was produced using QGIS 3.24.

## Results

### Description of the patients with Q fever

Over the 11 years of the study period, 695 patients (464 men (66.8%), 231 women (33.2%), M/F sex ratio of 2.0) were included (Fig 1). The median age was 45.3 years (IQR: 33.7–56.3) 45.3 (34.6–55.7) for men and 45.3 (33.2–58.1) and the majority (65.9%) belonged to the age groups between 30 to 59 years (Fig 2).

The acute form was by far the most frequent presentation (678 cases, 97.6%), with pneumonia being a common clinical picture (n = 526, 77.5%), followed by isolated fever (n = 113, 16.7%) (Fig 1). The details of the remaining clinical pictures are reported in Fig 1. In our study, only 17 persons out of 695 (2.4%) presented a persistent focalized form at the onset of the symptoms (some patients secondarily developed a persistent form during the follow-up of the acute Q fever and were not included as persistent form). Endocarditis was the most common (n = 12/17, 70%), followed by dilated cardiomyopathy (n = 3/17, 18%). No vascular infection was primary diagnosed.

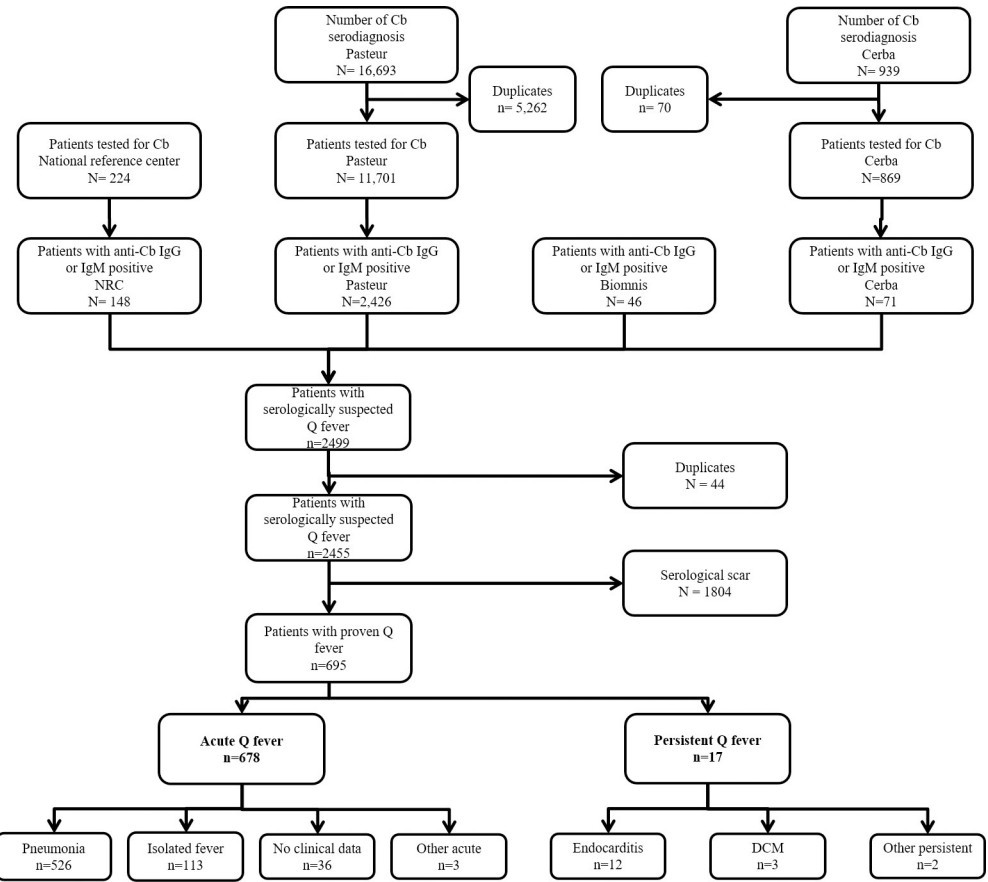

**Fig 1. Flowchart of the study. Details of the cases of Q fever between 2007 and 2017.** *Cb* = *Coxiella burnetii*; QF = Q fever; DCM = Dilated cardiomyopathy. Other acute: 1 asymptomatic seroconversion in a women tested in the assessment of her husband's family who had a symptomatic form; 1 acute meningitis, 1 acute myopericarditis. Other persistent: 1 infectious chronic polyarthritis; 1 granulomatous hepatitis in a patient with immunosuppressive treatment for rheumatoid arthritis.

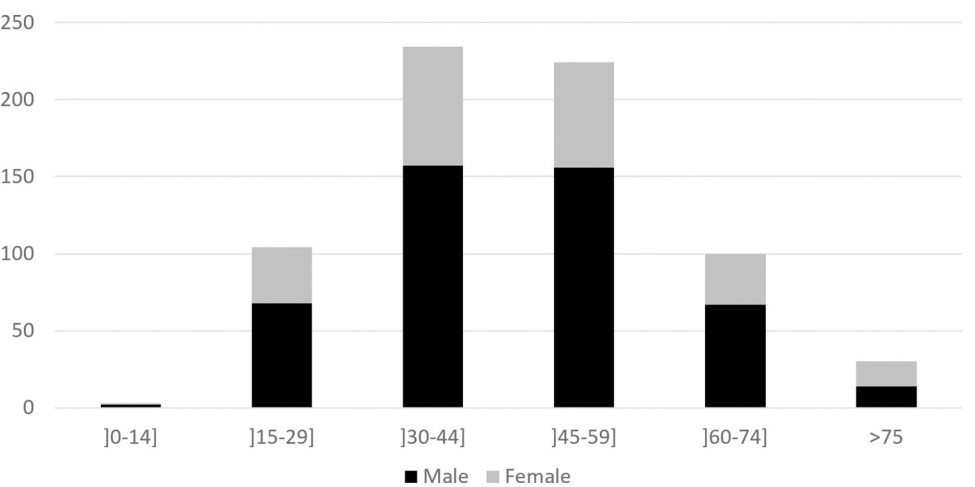

**Fig 2. Number of cases according to the age category.** F = female, M = male.

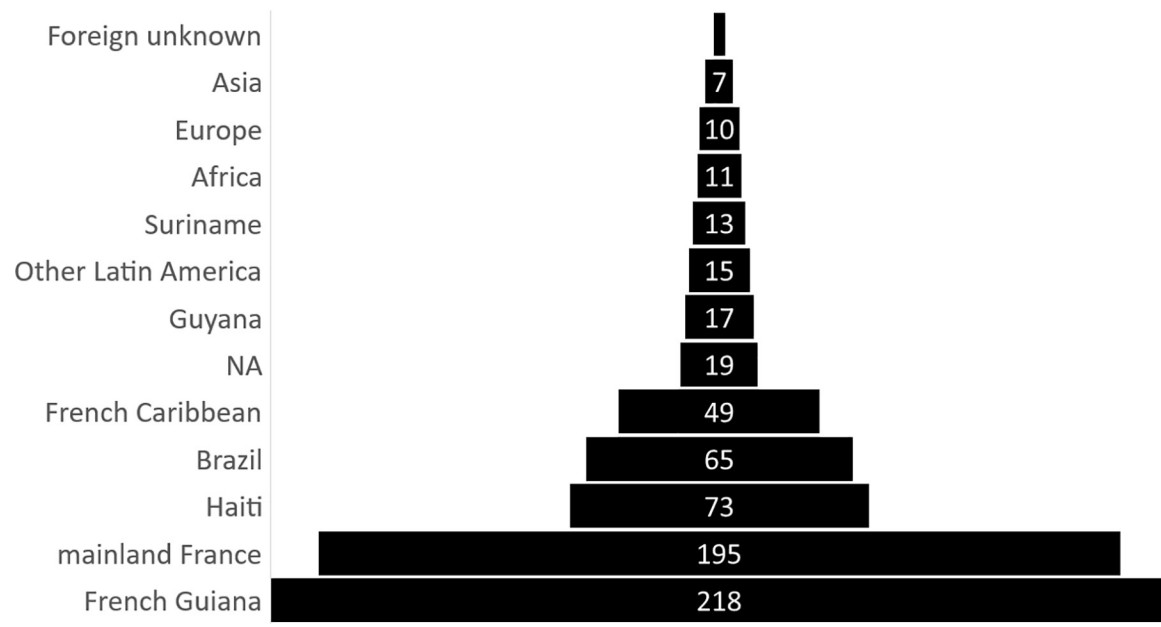

**Fig 3. Country or territory of birth of patients infected by Q fever 2007–2017.** NA: Country of birth unknown; Foreign unknown: not born in France, nor French Guiana or French Caribbean, but information of foreign country not available.

The countries of birth were French Guiana (n = 218, 31.4%), followed by mainland France (28.1%), Haiti (n = 73, 10.5%), Brazil (n = 65, 9.4%) and French Caribbean (n = 49, 7.1%), (Fig 3).

When looking more closely at the place of residence, we found that the 3 cities on the so-called "island of Cayenne" that gathered Cayenne, Rémire-Montjoly and Matoury, small peninsula separated of the rest of the continent by three rivers, totaled most of the cases (n = 592/689, 85.9%). Thus, they represent respectively Cayenne (n = 251/689, 36.1%), Rémire-Montjoly (n = 189/689, 27.4%), and Matoury (152/689, 22.0%) (Fig 4 and Table 1). If we add the 3 closest municipalities, Roura, Montsinéry-Tonnégrande and Macouria, they all together represent 643 among 689 identified place of residence (93.3%). In addition, we noticed a strikingly low number of patients from the western part of French Guiana: Saint Laurent du Maroni (n = 16), and Kourou (n = 4, Fig 5) while the population of these two cities represents up to 17% and 10%, respectively, of the French Guiana population according to the INSEE 2015 census (http://www.insee.fr). Forty per cent of the cases (282/695) and 59.5% (413/695) occurred during the dry season (Fig 6).

### Risk factors for Q fever

In a bivariate analysis, after comparison to the 2015 population census data, factors associated with *Cb* infection were: male gender (OR 2,02; (95%CI: 1,72–2,38), p<0.001), being born in mainland France (OR 2,96 (95%CI: 2,49–3,50), p<0.001), an age between 30 to 59 years old (OR 4,02 (95%CI 3,01–4,15), p<0.001) and a place of residence in Cayenne and surroundings (OR 6,05 (95%CI: 4,88–7,55), p<0.001) (Table 2).

### Calculation of the incidence rate

The QF annual incidence rate was 27.4 (95%CI: 7.1–47.7 per 100,000 inhabitants), ranging from 5.2 in 2007 to 40.4 in 2010 (Fig 7). The rate remained rather stable after 2010 at a level

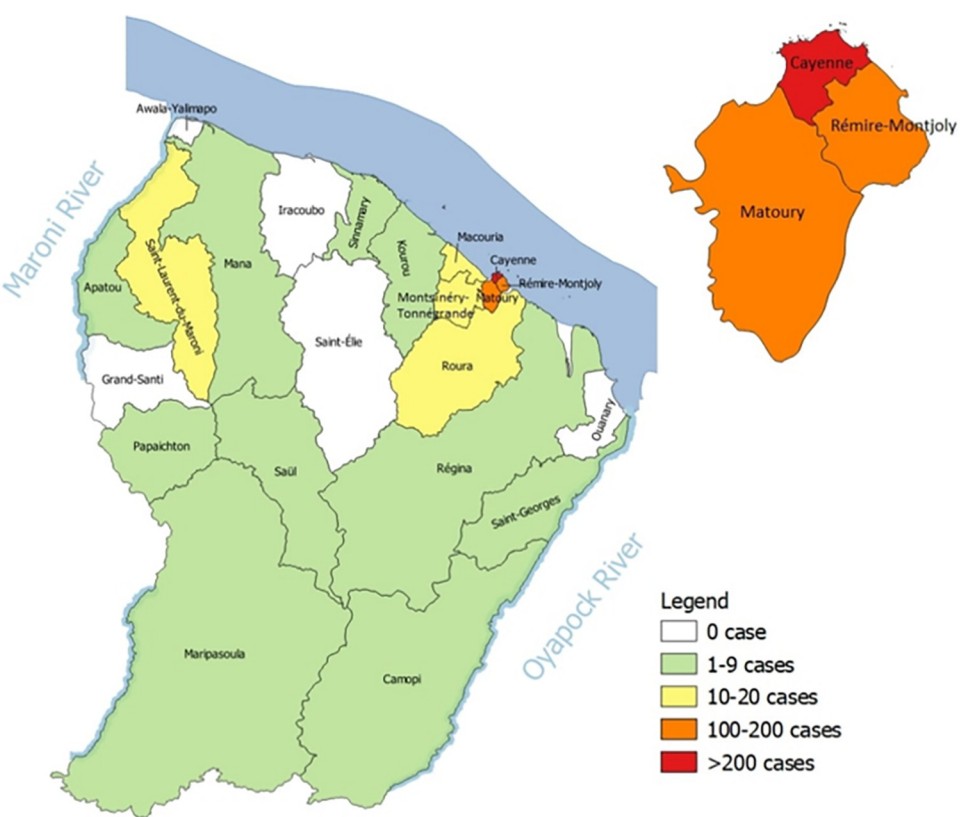

**Fig 4. Mapping of the cases of Q fever per municipality in French Guiana 2007–2017.** Source: QGIS software, release 3.24. Link to the base layer: https://www.insee.fr/fr/information/2028028.

around 30 cases per 100,000 inhabitants. The annual incidence between 1990 to 2017 is presented in Fig 7 showing the increase between 1996 to 2001, the peak in 2005 followed by a low incidence between 2006 to 2010 [15].

## Calculation of the positivity rate

For the calculation of the positivity rate, we secondarily contacted the 4 laboratories. This was not easy, because for several of them, the software had changed, and the number of tests performed over the study period as well as the number of patients having received a diagnostic test for *C. burnetii* could not be obtained from all of them, and over the entire study period (Table 3).

Thus, the number of patients with non-zero *C. burnetii* serology over the study period was 17.4% (N = 2426/11701) for the laboratory of the Institut Pasteur in French Guiana over the period 2007–2017, 66.1% (N = 148/224) for the national reference center in Marseille over the period 2014–2017, and 8.2% (N = 71/869) for the Cerba laboratory. Data could not be obtained retrospectively for the Biomnis laboratory, but a very limited number of tests had been sent to them over the period. Finally, if we relate the number of patients with a diagnosis of Q fever (acute or chronic) over the study period to the number of patients tested, we find a positivity rate of 5.6% and a number of patients with a non-zero *C. burnetii* serodiagnosis related to the total number of patients tested over the study period of 27.8% (695/2,499).

**Table 1. Number of cases on the 2007–2017 and compared to the population of each municipality.**

| | Municipality | Number of cases | Population | Ratio (%) |
|---|---|---|---|---|
| | Community of municipalities of the central coastline | | | |
| | Cayenne | 251 | 57,614 | 0.44 |
| | Macouria | 23 | 11,719 | 0.20 |
| | Matoury | 152 | 32,427 | 0.47 |
| | Montsinéry-Tonnégrande | 12 | 2,473 | 0.49 |
| | Rémire-Montjoly | 189 | 23,976 | 0.79 |
| | Roura | 16 | 3,713 | 0.43 |
| | Community of municipalities of savannahs | | | |
| | Iracoubo | 0 | 1,878 | 0 |
| | Kourou | 4 | 26,221 | 0.02 |
| | Saint Elie | 0 | 95 | 0 |
| | Sinnamary | 4 | 2,957 | 0.14 |
| | Community of municipalities of Eastern Guiana | | | |
| | Camopi | 2 | 1,769 | 0.11 |
| | Ouanary | 0 | 165 | 0 |
| | Régina | 5 | 946 | 0.53 |
| | Saint Georges de l'Oyapock | 2 | 4,020 | 0.50 |
| | Community of municipalities of Western Guiana | | | |
| | Apatou | 1 | 8,431 | 0.01 |
| | Awala-Yalimapo | 0 | 1,379 | 0 |
| | Grand-Santi | 0 | 6,969 | 0 |
| | Mana | 1 | 10,241 | 0.01 |
| | Maripasoula | 6 | 11,856 | 0.05 |
| | Papaïchton | 4 | 7,266 | 0.06 |
| | Saint Laurent du Maroni | 16 | 43,600 | 0.04 |
| | Saül | 1 | 150 | 0.67 |
| | NA | 6 | | |
| | Total | **695** | | |

NA: municipality of residence not known.

## Discussion

This study presented a very large sample of *Cb*-infected persons with 695 patients. In comparison, the national reference center found 3723 patients with acute q fever and 1675 with chronic q fever (previous designation of persistent form) between 1985 to 2009 in the whole mainland France, and 2434 cases on the 1991–2016 period, where the population is over 60 million of inhabitants, while around 270,000 inhabitants live in French Guiana [2,18]. The annual incidence of Q fever in French Guiana remains high, about 40 case/100,000 inhabitants per year, and seems to be stable since 2009, compared to previous years. It probably represents the highest incidence in the world. The mean of annual incidence between 1990 and 2007 was 29.9 cases per 100000 inhabitants but with large temporal variations: 0 case in 1995 versus120.7 cases per 100000 inhabitants in 2004(10). In this study, the incidence was stable between 2010 and 2017 around 30 per 100000 inhabitants. We did not observe an incidence as high as in 2005 (150cases per 100,000 inhabitants per year). Q fever incidence in French Guiana remains strikingly high and seems to be an exception: in mainland France, annual incidences of acute Q fever and endocarditis were 2.5/100000 persons and 0.1/100,000 persons, respectively between 1985 to 2009 [2]. These results confirmed previous published data by Edouard *et al.*

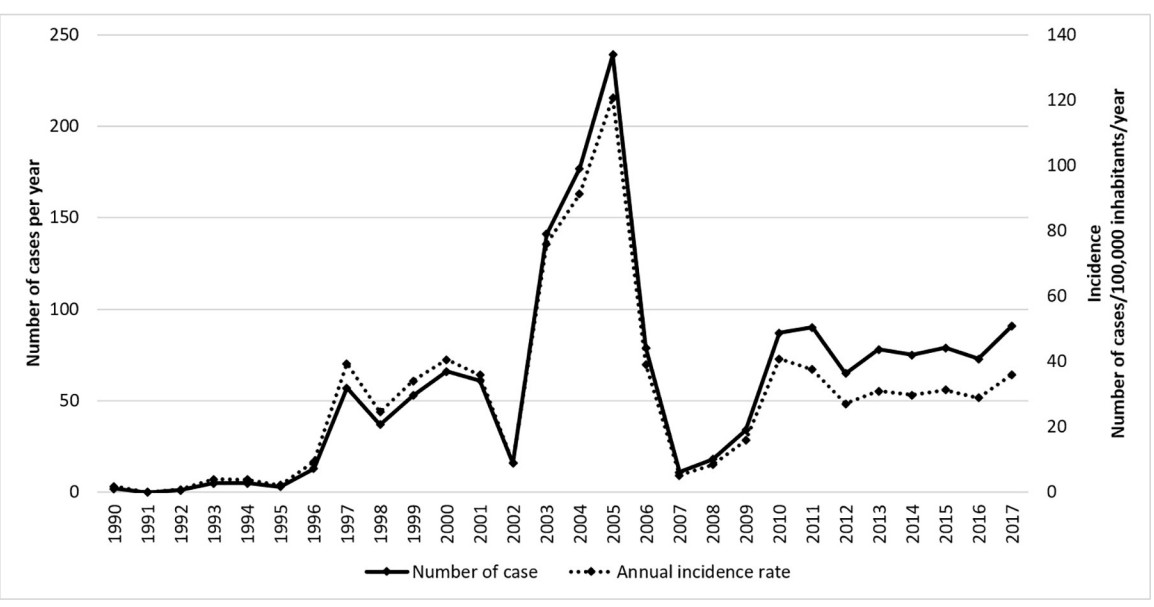

**Fig 5. Number of cases and annual incidence between 1990 and 2017.**

where acute Q fever incidence in Cayenne (17.5/100,000 persons) was significantly higher than in Marseille, France (1.9/100,000) [8]. There is no comparison of such a high incidence in Latin America or in the rest of the world, even in countries with a high incidence: for example, in Israel, where Q fever is endemic, incidence is between 1 to 2 to 100000 inhabitants per year [19,20]. In USA, incidence of Q fever was between 0.1 to 0.6 per million inhabitants per year between 2000 to 2012 [21].

The M/F sex ratio was 2.0, higher than for the 1950–2006 period (sex ratio at 1.3) [22]. The mean age was 45.3 years, and remained stable since 1950 (46.8 between 1950 to 2006).Male predominance in symptomatic acute Q fever is well described in the literature, and hormonal factors have been shown in animal models, but it remains to be fully elucidated [1,23].Whether some wild species such as three-tooth sloth (*Bradypus tridactylus*) and capybara (*Hydrochoerus hydrochaeris*) have been evidenced as potential reservoir of the bacterium, the epidemiological cycle remains unclear and no reservoir has been found in the usual domestic animals such as sheep, goats and cattle [24–27].The overrepresentation of the category of males between 30 to 60 years, may reflect a work-associated transmission as described for other zoonosis such as leptospirosis [28].

Acute forms were observed in 97.6% of cases vs. 69% in mainland France [2].The specific serogroup MST17 of French Guiana and its potential increased virulence may be incriminated [10,11]. Q fever serodiagnosis is performed quite systematically for community acquired pneumonias, as physicians are aware of the importance of Q fever in our territory [9]. This may explain the overrepresentation of acute forms in our study. Pneumonia was the most frequent clinical presentation. This particularity was previously observed and has therapeutical consequences: in fact *Cb* is implicated in 24% to 38% of pneumonias in the area of Cayenne, which is the highest prevalence ever described worldwide [9,29].Therefore, doxycycline is systematically included in empirical antibiotherapy of community acquired pneumonia in Cayenne.

In the present study, to be born in mainland France was a significative risk factor to be infected by *Coxiella burnetii* with an OR at 6.33. A residence in Cayenne area as a risk factor of Q fever has already been evidenced in previous studies [14,15,29]. These two phenomena

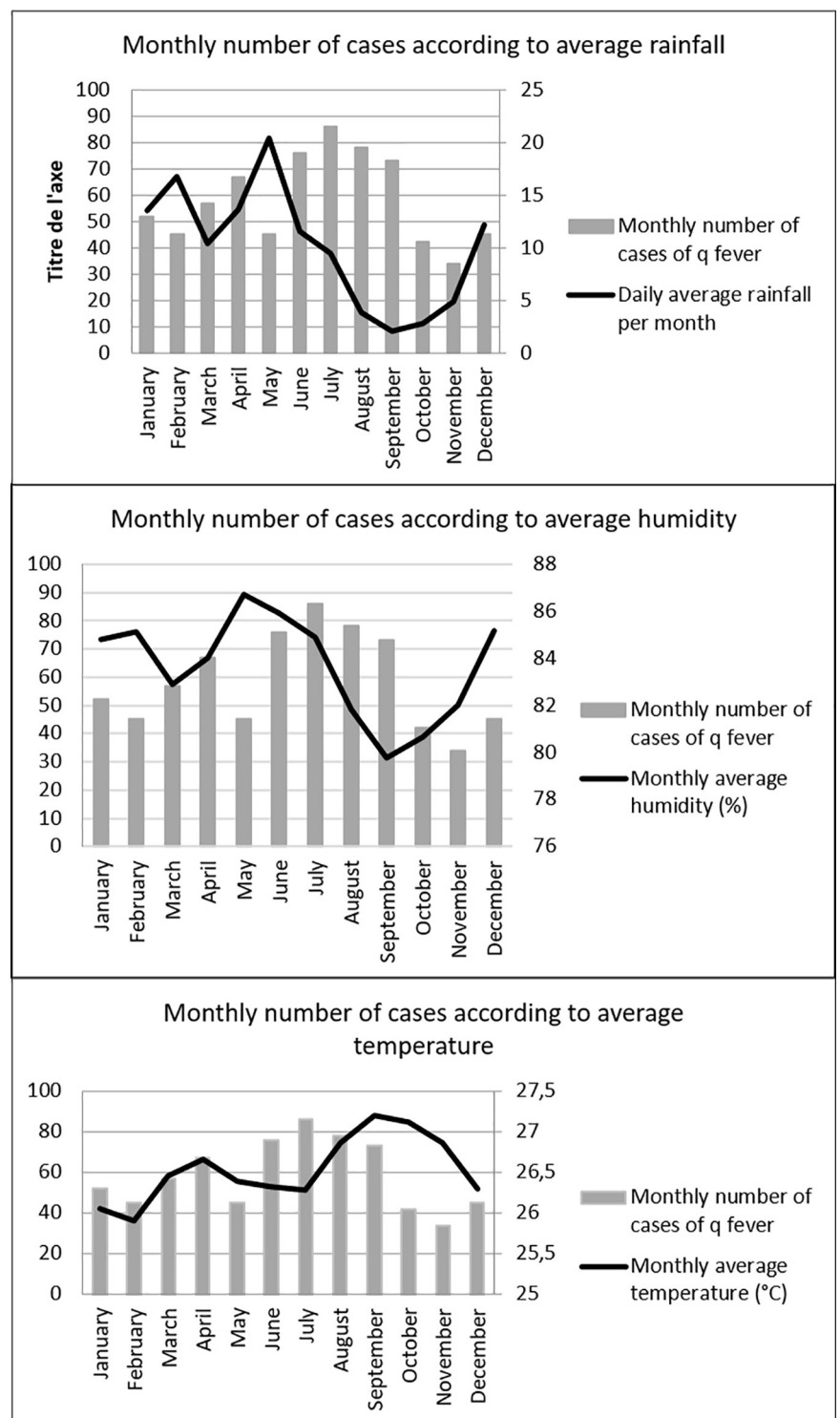

**Fig 6. Monthly distribution of the cases of Q fever between 2007 and 2017 according to three climate variables: rainfall, humidity and temperature.**

**Table 2. Determinants of patients with Q fever compared to general population in French Guiana, 2007–2017 (Source: Insee census 2015).**

| Risk factors | | Study population N(%) | General Population N(%) | OR (95%CI), p |
|---|---|---|---|---|
| **Age** | 30–59 y | 458 (65.9) | 91921 (35.4) | OR 4,02 (IC95% 3,01–4,15), <0.001 |
| | <30 or >59y | 237 (34.1) | 167944 (64.6) | |
| | total | 695 | 259865 | |
| **Gender** | Male | 464 (66.6) | 128973 | OR 2,02 (IC95% 1,72–2,38), <0.001 |
| | Female | 231 (33.4) | 130892 | |
| | total | 695 | 259865 | |
| **Origin** | European/Mainland France | 195 (28.8) | 28505 (12) | OR 2,96 (IC95% 2,49–3,50), <0.001 |
| | Other | 481 (71.2) | 209043 (88) | |
| | Total | 676 | 237549 | |
| **Country of birth** | France (including French Guiana & French West Indies) | 462 (68.3) | 154562 5 (65.1) | p = 0.9 |
| | Foreign | 214 (31.7) | 82987 (34.9) | |
| | Total | 676 | 237549 | |
| **Place of residence** | Area of Cayenne | 596 (86.0) | 125577 (49.8) | OR 6,05 (IC95% 4,88–7,55), <0,001 |
| | Rest of French Guiana | 97 (14.0) | 126761 (50.2) | |
| | Total | 693 | 252338 | |

which have already been found in previous study are not easy to understand [9,29]. Indeed, the main risk factors associated with infection with *Cb* were to work in the construction industry, in gardening, and in particular to pass the brush cutter and to have one's home close to the forest [14,25,27]. These criteria are not specific to the population originating from mainland France. On the contrary, the risk factors mentioned above rather concern populations resulting from more recent migrations, in particular from Haiti, Brazil, Suriname, which are however poorly represented here. On the other hand, the populations most affected in our study, namely the French citizens from French Guiana and from mainland France, possibly from a more affluent socio-economic situation, share the same privileged districts of Cayenne and its surroundings [30]. They also approximatively share the same way of life comparatively to other cultural groups. The hypothesis of genetic or

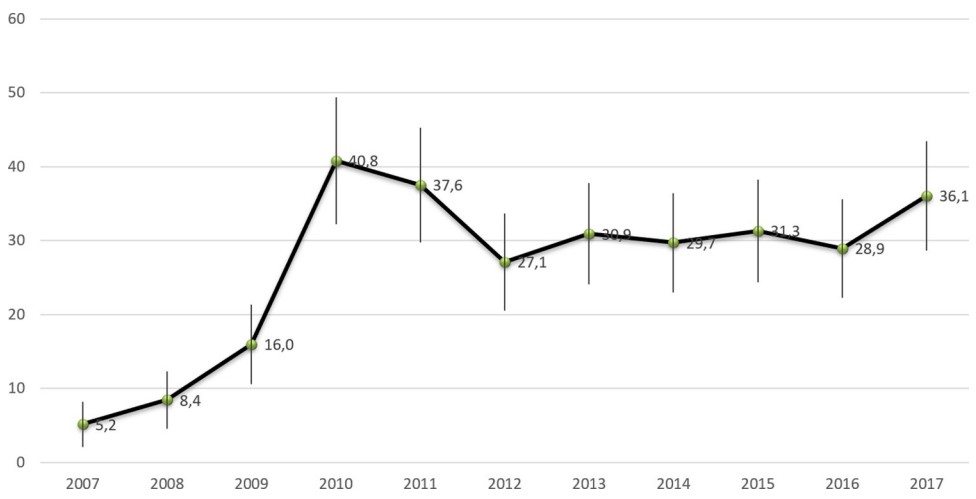

**Fig 7. Annual incidence of Q fever in French Guiana between 2007 and 2017.**

**Table 3. Rate of positivity according to the lab.**

| | Number of C. burnetii serodiagnosis performed | Number of patients with C. burnetii serodiagnosis | Number of patients with C. burnetii serodiagnosis and IgG or IgM superior to 0 | Positivity rate |
|---|---|---|---|---|
| **Pasteur 2007–2017** | 16,963 | 11,701 | 2,426 | 17.4% |
| **NRC 2014–2017** | NA | 224 | 148 | 66.1% |
| **Cerba 2007–2014** | 939 | 869 | 71 | 8.2% |
| **Biomnis 2007 2017** | NA | NA | 46 | NA |
| **Total** | 17,902 | 12,502 | 2,499 | 20.0% |
| **Diagnosis of Q fever** | | 695/12,502 = 5.6% | 695/2,499 = 27.8% | |

Pasteur = Institut Pasteur in French Guiana; NRC = National Reference center of Rickettssia and Q fever, Marseille; Biomnis = Biomnis laboratory, Ivry-sur-Seine, France; Cerba = Cerba, Saint-Ouen-l'Aumône, France

NA = not available any more (too old software)

ethnic predisposition seems to be refuted by the fact that the Creole people with African roots are also concerned. An ancillary seroprevalence study carried out throughout French Guiana initially dedicated to the seroprevalence of arboviruses and which was secondarily interested in Q fever, not yet published but presented in 2020 at a regional congress, showed conversely that the one of the most prevalent populations were migrants from Haiti, known to be one of the cultural groups most involved in gardening [31]. Furthermore, in the same seroprevalence study, they showed that high seroprevalence rates are found in some cities where Q fever cases are described, such as Maripasoula, a town located on the western border of French Guiana with Suriname. This contrast between results of seroprevalence studies and incidence studies remains to be elucidated. Several hypotheses could explain the high incidence in the Cayenne area: 1.the Institut Pasteur in French Guiana, where most of the serodiagnosis are performed is located in Cayenne; 2. the clinicians are more aware of Q fever in Cayenne area. Indeed, a thesis of medicine not published showed that general practitioners (GP) from Cayenne and surrounding had significative better knowledge of Q fever than the other GP from the rest of FG [32]. Subsequently, the serodiagnosis of *Cb* is more rarely performed in St Laurent du Maroni, and Kourou than in Cayenne. Furthermore, most of the other cities are located in remote areas, and lab tests are less easily available. At last, an immunological hypothesis where people from French Guiana have protective immunity due to the hyperendemic mode of QF, and present asymptomatic forms of QF whereas newly arrived persons may not have this protective immunity. In conclusion, although there are several tentative hypotheses, we lack information about animal reservoirs, immunity, transmission route, and many questions remain unanswered for the moment.

We observed that cases of Q fever were more frequent during dry season (415 persons) than during raining season (285 persons). It can be explainable by the fact that dry season is more favorable for aerosolization of the bacteria contained in dust. There was a delay of two months between the rain peak and cases of Q fever, as already described previously [33]. And the peak of cases is 2 months before the peak of temperature. Gardon *et al.*in contrast observed a strong correlation between rainfall and Q fever incidence. In their study, the incidence seemed to be maximal during the raining season. They hypothesized that a potential wild reservoir had a rain-dependent activity as well as another study [14,33].

## Conclusion

The endemicity of Q fever is high and stable in French Guiana and the incidence is the highest in the world. It may be considered as an hyperendemic region for Q fever. Several factors associated to *Cb* infection have been identified: being a man, being born in mainland France, being aged between 30 to 59 years and residing in Cayenne and surroundings. But the physiological reasons of these associations are not well understood. Additional studies are needed to further understand the drivers of this infection that has barely been described in the rest of South America.

## Acknowledgments

The authors thank Magali Dodemont (Institut Pasteur in French Guiana), Anne Ovize (Biomnis laboratory), Sabine Trombert (Cerba laboratory) and Léa Luciani (NRC of Ricketssiae and Q fever) for the new extraction of the data. They also thank Sébastien Rabier for the realization of the map of French Guiana, and jean-Michel Cauvin, Milko Sobesky and Ward Schrooten for kindly providing data from the medicalization of information systems program of the hospital of Cayenne. At last, authors thank the following colleagues to have supported the patients: (alphabetic order)

Philippe Abboud, Bastien Bidaud, Sylvie Bisser, Timothée Bonifay, Jérémie Bouche, Anne-Marie Bourbigot, Mathilde Boutrou, Muriel Chiron, Pierre Couppié, Franck De Laval, Pierre Douat, Fabien Duigou, Mélanie Gaillet, Geneviève Guillot, Stéphanie Houcke, Hatem Kallel, Franck-Yves Kénol, Valérie Leissing, Dominique Louvel, Aba Mahamat, Enguerrane Martinez-Lorenzi, Alessia Melzani, Céline Michaud, Emilie Mosnier, Patric Mulhausen, Elise Ouedraogo, Marie-Claire Parriault, Agathe Pastre,Vincent Pommier de Santi, Valérie Pouzyreff, Laure Slugacz, Stéphanie Thomas, Guillaume Velut, Guillaume Vesin, Gaëlle Walter

## Author Contributions

**Conceptualization:** Pauline Thill, Loïc Epelboin.

**Data curation:** Carole Eldin, Laureen Dahuron, Alain Berlioz-Artaud, Emmanuel Beillard, Loïc Epelboin.

**Formal analysis:** Emmanuel Beillard, Loïc Epelboin.

**Investigation:** Loïc Epelboin.

**Methodology:** Mathieu Nacher, Emmanuel Beillard, Loïc Epelboin.

**Supervision:** Mathieu Nacher.

**Validation:** Mathieu Nacher, Félix Djossou.

**Writing – original draft:** Pauline Thill, Loïc Epelboin.

**Writing – review & editing:** Carole Eldin, Alain Berlioz-Artaud, Magalie Demar, Mathieu Nacher, Emmanuel Beillard, Félix Djossou.

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
