## [Decision Letter · Decision Letter 0]

3 Oct 2021

Dear Dr. Epelboin,

Thank you very much for submitting your manuscript "High endemicity of Q fever in French Guiana: a cross sectional study (2007-2017)" for consideration at PLOS Neglected Tropical Diseases. As with all papers reviewed by the journal, your manuscript was reviewed by members of the editorial board and by several independent reviewers. In light of the reviews (below this email), we would like to invite the resubmission of a significantly-revised version that takes into account the reviewers' comments. 

We cannot make any decision about publication until we have seen the revised manuscript and your response to the reviewers' comments. Your revised manuscript is also likely to be sent to reviewers for further evaluation.

Sincerely,

Wen-Ping Guo

Associate Editor

Javier Pizarro-Cerda

Deputy Editor

Reviewer's Responses to Questions

**Key Review Criteria Required for Acceptance?**

**Methods**

-Are the objectives of the study clearly articulated with a clear testable hypothesis stated?

-Is the study design appropriate to address the stated objectives?

-Is the population clearly described and appropriate for the hypothesis being tested?

-Is the sample size sufficient to ensure adequate power to address the hypothesis being tested?

-Were correct statistical analysis used to support conclusions?

-Are there concerns about ethical or regulatory requirements being met?

Reviewer #1: while important background information is given in the introduction certain elements are missing. There should be a brief description of the medical resources available in French Guiana - how many doctors? Has the health department informed doctors about the high rate of Q fever in FG in the past?

How many serum samples are tested each year in FG - give the number and percent positive per year.

Biominis or Cerba Central labs -- where are they located. 

Did all 4 laboratories that performed tests for C. burnetii use the same materials - eg did one supplier provide the materials to all 4 labs? 

Did all 4 labs use the same criteria for a positive test.

Fischers test -- ?? Fishers test

Reviewer #2: (No Response)

Reviewer #3: Cite the references showing that the criteria for serodiagnosis of Q fever are appropriate: Persistent Q fever generally has higher antibody titers to Phase I than to Phase II.

Describe in more detail the epidemiological items and risk factors investigated.

Even if the analysis is based on anonymization, the ethical response is insufficient. It should be stated that the research plan has been approved by an authorized committee. The method of obtaining information from the subject by telephone may be inappropriate in many cases as informed consent.

**Results**

-Does the analysis presented match the analysis plan?

-Are the results clearly and completely presented?

-Are the figures (Tables, Images) of sufficient quality for clarity?

Reviewer #1: You indicated that you would exclude patients if data were missing - the results do not indicate that any patients were excluded.

It is important to give the percentage of serum samples that were positive each year

Of the 681 patients with acute Q fever - 513 had pneumonia and 22 others had a variety of illnesses - what about the 46 others??

Reviewer #2: (No Response)

Reviewer #3: After describing Method in more detail, show considerations in terms of race and other indicators of genetic background rather than country of birth. The median age should be shown by gender as well as overall.

For each city or region that you have categorized, show the ratio of the population by industry and the rate of positivity among them.

**Conclusions**

-Are the conclusions supported by the data presented?

-Are the limitations of analysis clearly described?

-Do the authors discuss how these data can be helpful to advance our understanding of the topic under study?

-Is public health relevance addressed?

Reviewer #1: You indicate that there is some uncertainty as to the source of Q fever in FG. References 20 and 21 are highly suggestive that the three toed sloth and Capybara play a role. Can you expand on the reason for your uncertainty.

Reviewer #2: (No Response)

Reviewer #3: Conclusion is simple and clear. However, Methods and Results are not enough.

So, this is a weak basis for consideration.

**Editorial and Data Presentation Modifications?**

Reviewer #1: (No Response)

Reviewer #2: (No Response)

Reviewer #3: No

**Summary and General Comments**

Reviewer #1: This is an important paper

Reviewer #2: The authors concluded that the incidence of Q fever in French Guiana remained high and stable and the highest in the world. They also found that the risk factors associated with Q fever were male gender, being born in mainland France with age between 30 to 59 years-old and a residence in Cayenne and surroundings.

The description of method is too simple and rough. What is the method for detecting antibodies in patients with Q fever? Is it a commercial kit or a homemade one? How specific is this reagent? When these problems are not clear, it is impossible to judge whether these patients really have Q fever. The author mentioned PCR in the method. What are the primers and protocol of the PCR? Is any patient PCR positive? What evidence to suppot the claim that the incidence of Q fever in French Guiana is the highest in the world. In addition, misspellings such as acutez Q fever and inonly in hte manuscript need to be corrected.

Reviewer #3: Updating epidemiological information in endemic areas with high Q fever is extremely important and meaningful. However, the description of Methods and Results is insufficient, and many of the discussions are based on multiple unpublished data. I would like to see the presentation of more detailed information and a discussion that eliminates unpublished data as much as possible.

Also carefully present climatic and geographical information, even if it is a secondary objective.　Only a small percentage of readers of international journals are familiar with French Guiana.

PLOS authors have the option to publish the peer review history of their article (what does this mean?). If published, this will include your full peer review and any attached files.

Reviewer #1: Yes: Thomas J. Marrie MD

Reviewer #2: No

Reviewer #3: No
---

## [Decision Letter · Decision Letter 1]

22 Mar 2022

Dear Dr. Epelboin,

We are pleased to inform you that your manuscript 'High endemicity of Q fever in French Guiana: a cross sectional study (2007-2017)' has been provisionally accepted for publication in PLOS Neglected Tropical Diseases.

Best regards,

Wen-Ping Guo

Associate Editor

Javier Pizarro-Cerda

Deputy Editor

Reviewer's Responses to Questions

**Key Review Criteria Required for Acceptance?**

**Methods**

-Are the objectives of the study clearly articulated with a clear testable hypothesis stated?

-Is the study design appropriate to address the stated objectives?

-Is the population clearly described and appropriate for the hypothesis being tested?

-Is the sample size sufficient to ensure adequate power to address the hypothesis being tested?

-Were correct statistical analysis used to support conclusions?

-Are there concerns about ethical or regulatory requirements being met?

Reviewer #1: this is a revision . the authors have addressed all the questions i raised satisfactorily.

**Results**

-Does the analysis presented match the analysis plan?

-Are the results clearly and completely presented?

-Are the figures (Tables, Images) of sufficient quality for clarity?

Reviewer #1: see above

**Conclusions**

-Are the conclusions supported by the data presented?

-Are the limitations of analysis clearly described?

-Do the authors discuss how these data can be helpful to advance our understanding of the topic under study?

-Is public health relevance addressed?

Reviewer #1: see above

**Editorial and Data Presentation Modifications?**

Reviewer #1: see above

**Summary and General Comments**

Reviewer #1: see above

PLOS authors have the option to publish the peer review history of their article (what does this mean?). If published, this will include your full peer review and any attached files.

Reviewer #1: **Yes: **thomas j marrie

---

## [Editor Report · Acceptance letter]

13 May 2022

Dear Dr. Epelboin,

We are delighted to inform you that your manuscript, "High endemicity of Q fever in French Guiana: a cross sectional study (2007-2017)," has been formally accepted for publication in PLOS Neglected Tropical Diseases.

Best regards,

Shaden Kamhawi

co-Editor-in-Chief

Paul Brindley

co-Editor-in-Chief
